# Reproductive Issues in Long-Term Surviving Patients following Therapy for Hodgkin's Disease in the Republic of North Macedonia: Risks of Infertility According to First-Line Treatment Regimens

Gazmend Amzai *, Oliver Karanfilski, Sonja Genadieva Stavrikj and Aleksandar Stojanovikj

University Clinic for Hematology, Medical Faculty, University "Sts. Cyril and Methodius", 1000 Skopje, North Macedonia; karanfilski@yahoo.com (O.K.); sgenstav@yahoo.com (S.G.S.); profstojanovic@yahoo.com (A.S.)
* Correspondence: dr.gazmend_amzai@hotmail.com; Tel.: +389-70-251082

**Abstract:** Infertility as a consequence of therapy presents a high psychosocial burden for HL patients. In the cohort of our analyzed patients, within the post-ABVD surviving patients, alterations of the spermogram were documented in a total of 6.1% of the male patients and 5.4% of the female patients developed amenorrhea. On the other hand, within the subgroup of surviving patients following BEACOPP chemotherapy, 60% of the male patients manifested defects in their spermogram, and as high as 28.6% of the female survivors reported loss of their monthly cycle. It has been reported on several occasions that even prior to treatment, the sperm of HL patients manifests poorer quality characteristics when analyzed against control specimens from healthy male donors. The analyzed results in ABVD-treated male HL patients confirm ABVD to be a safe regimen for males of all age categories, as well as for female patients under the age of thirty. In women above the age of 30, the infertility risk rate is relatively low (14%), which leaves the decision of preserving fertility to themselves. For all BEACOPP-treated female, as well as male patients, a consult with a reproductive medicine specialist is warranted prior to therapy, due to the high infertility risk, and the final decision should be made on an individual basis.

**Keywords:** Hodgkin's lymphoma; late complications; fertility; male; female

## 1. Introduction

Contemporary advances in individualizing therapeutic approaches have significantly improved outcomes in HL patients. Nevertheless, the issue of disconcerting possible adverse effects, resulting from treatment-related aspects, remain of critical concern. Using contemporary treatment regimens administered to HL patients, more than 80% of these can be cured (clearly recognizing the critical influence of the clinical stage of the disease and the established risk categories). However, these encouraging treatment options do also carry a risk of developing secondary infertility [1]. They can compromise the potential for having biological progeny in these patients, which is an issue of fundamental significance regarding the quality of life in surviving patients. Roughly speaking, nearly 50% of patients diagnosed with Hodgkin's lymphoma are under the age of 35 years, a large proportion of them not having come to a final decision regarding their marital status or prospects, which is why the issue of infertility after treatment presents a particularly high psychosocial burden in survivors of Hodgkin's lymphoma [2].

According to a Cleveland study, out of 283 younger patients treated for cancer, having survived a long period of time, but being childless until then, 76% wish to and are planning to have children. Among them, 43% express anxiety and distress since they feel a lack of reliable and veritable information regarding eventual late adverse effects of treatment on

their reproductive capacity, understanding that this issue encompasses various medical, social and ethical aspects [3]. Increasing interest and progress in this field led to establishing the term *oncofertility*, representing an area of concern for both biomedical and social science field authorities. It studies all aspects related to fertility options for the patient, choice of procedures and definition of objectives, with respect to assessment of cancer diagnosis, treatment and quality of life [4].

The extent of compromising fertility is influenced by several risk factors [5]:

− Individual patient characteristics, such as gender, age and state of fertility prior to treatment;
− Produced chemotherapy and radiotherapy effects, which are dependent on the type of medication, the size and location of the radiation field, the dose, duration and route of administration;
− The disease itself, since infertility in males can be a direct consequence of the cancer (primary testicular HL), or influenced by anatomical irregularities (retrograde ejaculation), whereas in females anatomical or vascular abnormalities of the uterus, imposed by the disease itself, can disallow natural pregnancy [6].

Different drugs can have different effects on patient fertility. Alkylating agents (cyclophosphamide, procarbazine, melphalan, chlorambucil, busulfan) are remarkably linked to infertility in both men and women. The therapy-induced infertility rates in Hodgkin's lymphoma patients were significantly different among patients undergoing chemotherapy according to the two most commonly used induction regimens, ABVD (doxorubicin/bleomycin/vinblastine/dacarbazine) and BEACOPP (bleomycin/etoposide/ doxorubicin/cyclophosphamide/vincristine/procarbazine/prednisone). Among the regimens used in HL patients, only ABVD is a combination without an alkylating agent, making it less gonadotoxic, i.e., causing virtually no permanent azoospermia in men (even though, according to literature reports, this rate varies from 0 to 4%). Additionally, secondary amenorrhea is reported in 3.9% of the female patients treated with ABVD. Infertility is statistically considerably more frequent in patients treated with the escalated BEACOPP regimen, reaching around 90% in male and around 51% in female patients [7–10].

A very important issue in women receiving anti-cancer treatment is the estimation of the capacity of the ovaries to maintain and re-establish their function (reserve) following therapy, which critically mirrors the infertility risk. Although not all female patients experience cycle cessation, even transient loss of the period strongly implies a state of future infertility. It is clear that fertility is not always restored solely by recovery of the period, but that other morphological and functional issues need to be addressed as well. A functional cycle is clearly a prerequisite for fertility, but an additional burden to the duration of this "window" in female patients treated for cancer is the observation that menopause occurs earlier in such women [11].

The capacity to preserve fertility, preserve or store gametes, embryos or gonadal tissue with various methods, has become a key issue to patients with HL treated with radiotherapy and chemotherapy, who become long-term survivors. Several options are available, including embryo cryopreservation, sperm cryopreservation, cryopreservation of ovarian tissue, cryopreservation of mature oocytes, ovarian suppression with GnRH (gonadotropin-releasing hormone) analogues, etc. It is necessary to determine which method is adequate for each individual patient, depending on the age of the patient, existence of a dedicated partner or not, as well as on the aspirations of the patient [11–17].

## 2. Materials and Methods

We have analyzed 287 patients with Hodgkin's lymphoma, diagnosed and treated at the University Clinic for Hematology in Skopje, in the period from 2005 until 2015 (Table 1). In order for the patients to be included in the study, it was necessary for them to meet the inclusion criteria of being ≥14 years of age and to have a documented histopathology diagnosis of Hodgkin's lymphoma. Each patient had been assessed in a standard manner, including a detailed disease history, insisting on potential specific data of interest, presence of symptoms characteristic for defining the B type of HL, marital status, how many children

they had before chemotherapy, for the women in their reproductive age regarding the regularity of their cycle before and following treatment and for the male patients an assessment of their spermogram, in order to evaluate the fertility potential before and after treatment. Following a minimum of three days with no sexual activities, semen samples were obtained by the patients evaluated in the participating centers. The semen samples were analyzed for health and viability of sperm (sperm volume, sperm count, morphologic criteria of spermatozoa and sperm forward motility) according to World Health Organization (WHO) guidelines [18].

**Table 1.** Patient characteristics.

| Characteristic | | No (%) |
|---|---|---|
| Gender | Male | 152 (53%) |
| | Female | 135 (47%) |
| Age | Median age, years | 36.6 |
| | <35 years | 152 (52.9%) |
| | >35 years | 135 (47.1%) |
| Pathology Subtype | Nodular sclerosis | 131 (45.6%) |
| | Mixed cellularity | 90 (31.4%) |
| | Lymphocyte depletion | 10 (3.5%) |
| | Lymphocyte predominant | 26 (9.1%) |
| | Nodular lymphocyte predominant | 17 (5.9%) |
| | Nondifferentiated | 13 (4.5%) |
| Clinical Stage | I | 49 (17%) |
| | II | 95 (33.1%) |
| | III | 64 (22.3%) |
| | IV | 74 (25.8%) |
| | Undefined | 5 (1.8%) |
| Characteristic symptoms | Yes (B type) | 62.70% |
| | No (A type) | 37.30% |

The median observation period for the analyzed population is just above ten years. The long-term concept is to continue follow-up and periodical analyses for this group of patients every three years. Another important moment is that we plan to introduce additional contemporary and innovative procedures and methods in the assessment of infertility, as they become available and receive confirmation as being beneficial.

Two treatment groups were created: male and female survivor patients with Hodgkin's lymphoma, further subdivided in patients treated with ABVD and patients treated with high gonadotoxic chemotherapy BEACOPP. The purpose of this study was to evaluate the variations and alterations in fertility in distinct patient subgroups, according to initially administered treatment options.

Informed consent was obtained in accordance with the Declaration of Helsinki.

In order for a patient to become eligible for fertility status assessment in this study, a minimum of one-year disease-free period following treatment, i.e., complete remission, was a requirement.

Statistics: Statistical analyses were performed using the statistic program SPSS 17.0. The data were analyzed using the following statistical methods: chi-square, unpaired *t* test, Pearson's correlation test and Mann–Whitney test; the *p* value <0.05 was considered to depict statistical significance.

Associations between different variables and the fertility status were evaluated by logistic regression analyses. Demographics and disease characteristics are presented and summarized using descriptive statistics.

## 3. Results

### 3.1. Male Survivors

The total number of spermograms performed in the HD population was seventy.

Spermogram prior to chemotherapy was performed in 48 of the HL patients who had complete results available for statistical evaluation. Among them, normal findings were reported in 85.4% (41 pts), while in 14.6% (7 pts) of the patients, abnormalities in the spermogram were found. Asthenoteratozoospermia, oligozoospermia or azoospermia were documented in 4.2% of these patients. In one patient, oligoasthenozoospermia was found (Figure 1).

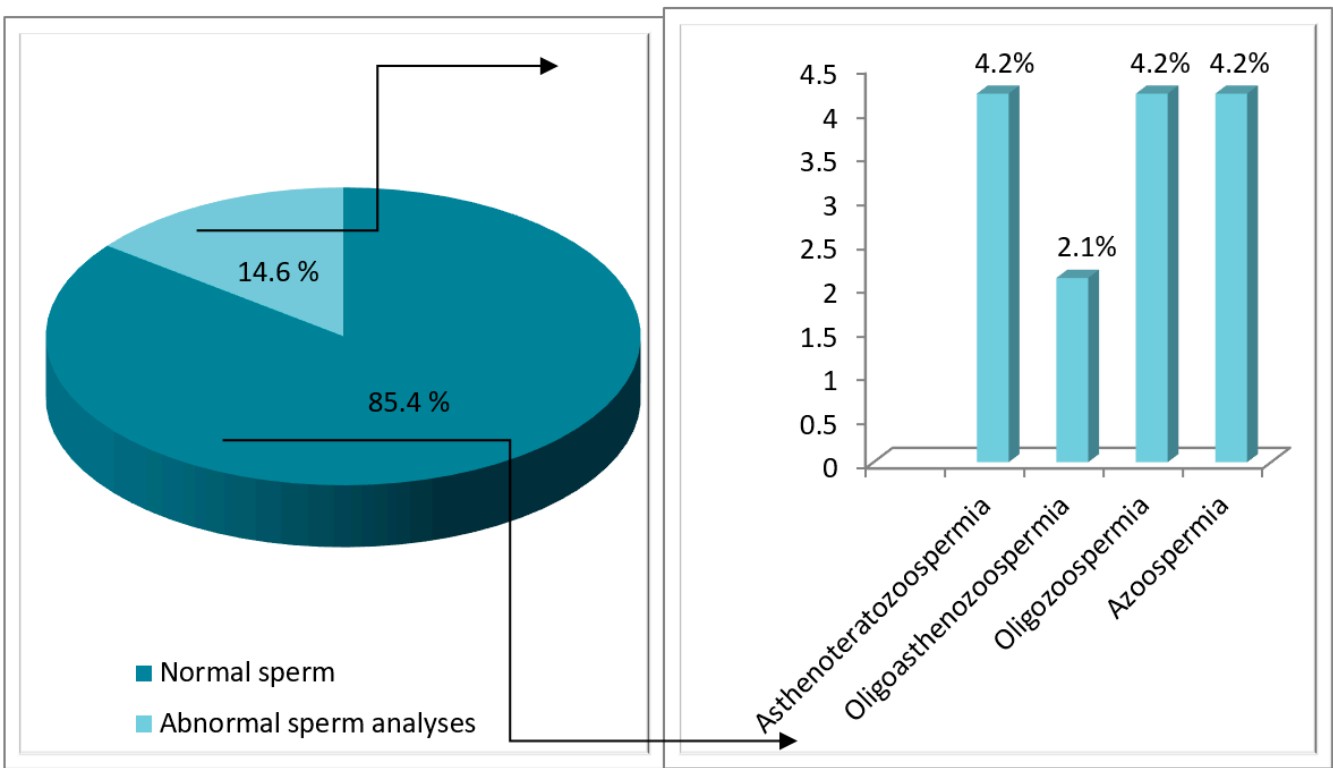

**Figure 1.** Distribution of HD patients according to spermogram findings prior to treatment.

A statistically significant correlation is registered between spermogram findings (normal and pathological) and the initial erythrocyte sedimentation rate (ESR) values below and over 50 (Pearson chi-square= 8.58261, *p* = 0.003394), showing that 85% of the patients with an abnormality in the spermogram before treatment also had elevated ESRs above 50.

Additionally, spermogram findings (normal and pathological) were documented to be statistically significantly correlated to the B type (with characteristic symptoms) of HD (Pearson chi-square= 10.1543, *p* = 0.006238), documenting that all patients with spermogram abnormalities prior to treatment were those manifesting constitutive HD symptoms (B type).

With regard to the clinical presentation of the hematological disease, aside from manifesting characteristic symptoms or not, we have analyzed our patients for possible correlations with the extent of clinical progression, i.e., disease (sites) involvement. Analyses were performed for single stages and for early vs. advanced disease stages as grouped data. Prior to any treatment, 41 of the 48 patients had a normal result for the examined spermogram, distributed as follows according to clinical stage of the disease: 10 in clinical

stage (CS) I, 16 in CS II, 10 in CS III and 5 in CS IV. Following ABVD chemotherapy, seven patients developed abnormalities in the spermogram, with the following CS distribution: none in CS I, two in CS II, one in CS III, three in CS IV and one in a patient lacking precise clinical stage determination. The employed statistical analyses did not succeed in revealing any statistically significant kind of correlation between the disease extent and occurrence of fertility abnormalities in this population, both when analyzed by single stages or when grouped as limited vs. advanced disease stages (*p* values falling in the upper ranges, >0.50, for all analyses). The lack of correlation persists both before treatment, as well as regarding alterations following treatment, although it needs to be underlined that patients with advanced stages were more often treated with the more aggressive regimen(s). Thus, even though the situation prior to treatment shows some correlations (ESR and B type of disease), the only impact on developing infertility indicators following treatment, were the chemotherapy regimens administered.

Following therapy, a spermogram was performed in 49 HD patients treated with ABVD. This group had an average age of 24.8 years, ranging from 15 to 43 years. In 93.9% (46 pts) of these patients, the spermogram result was normal, and 6.1% (3 pts) of patients following treatment with ABVD developed abnormalities. Teratozoospermia was documented in 4.1% of the patients and oligoasthenoteratozoospermia in one patient (Figure 2).

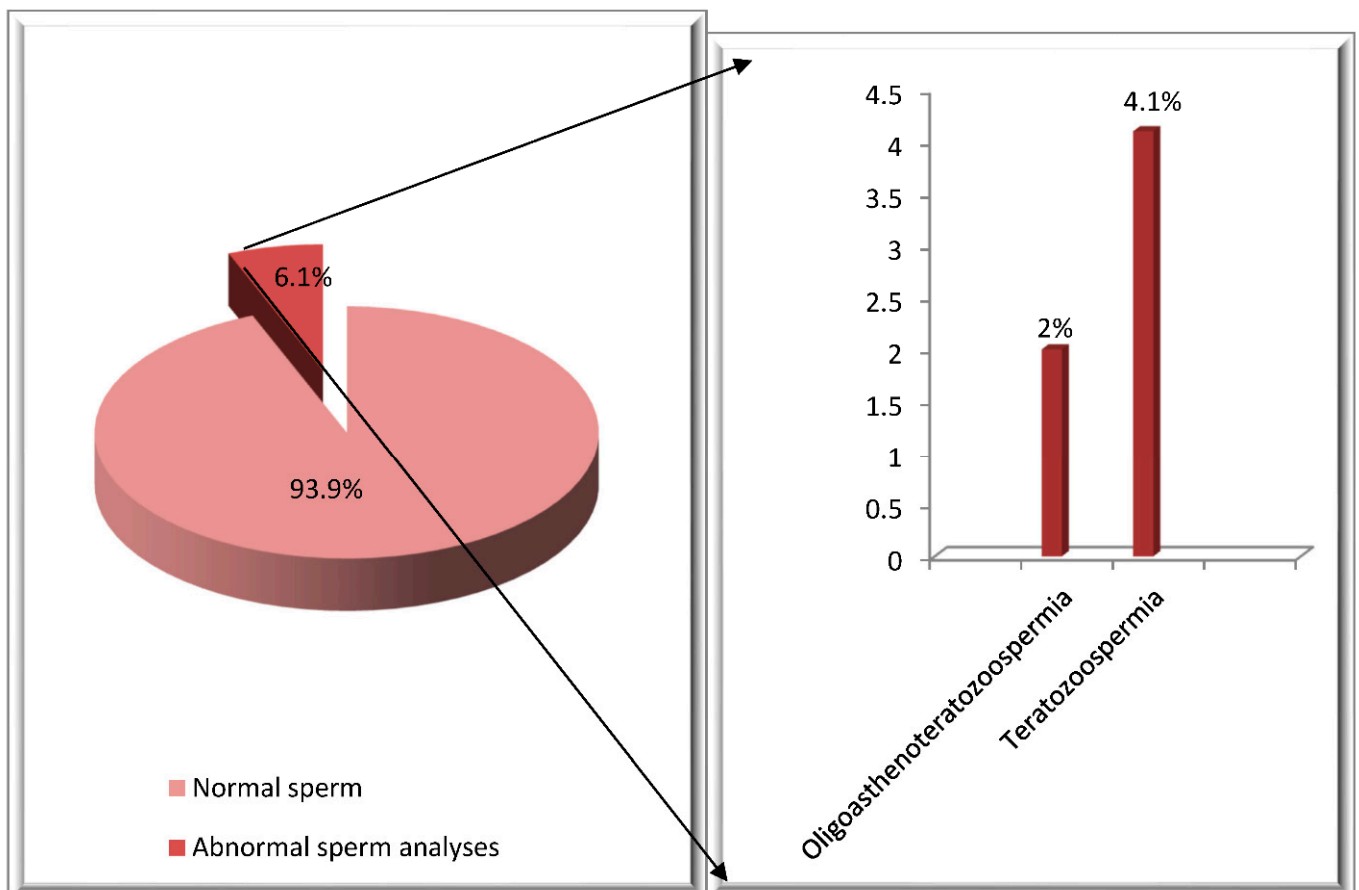

**Figure 2.** Distribution of HD patients according to spermogram findings following ABVD treatment.

Following BEACOPP treatment, a spermogram was performed in 10 HD patients, with an average age of 29.2 (range 19–38) years.

Among these patients, 40% (4 pts) had normal findings, while 60% (6 pts) manifested abnormalities following treatment. In five of these patients, azoospermia was documented, while one patient had astenozoospermia (Figure 3).

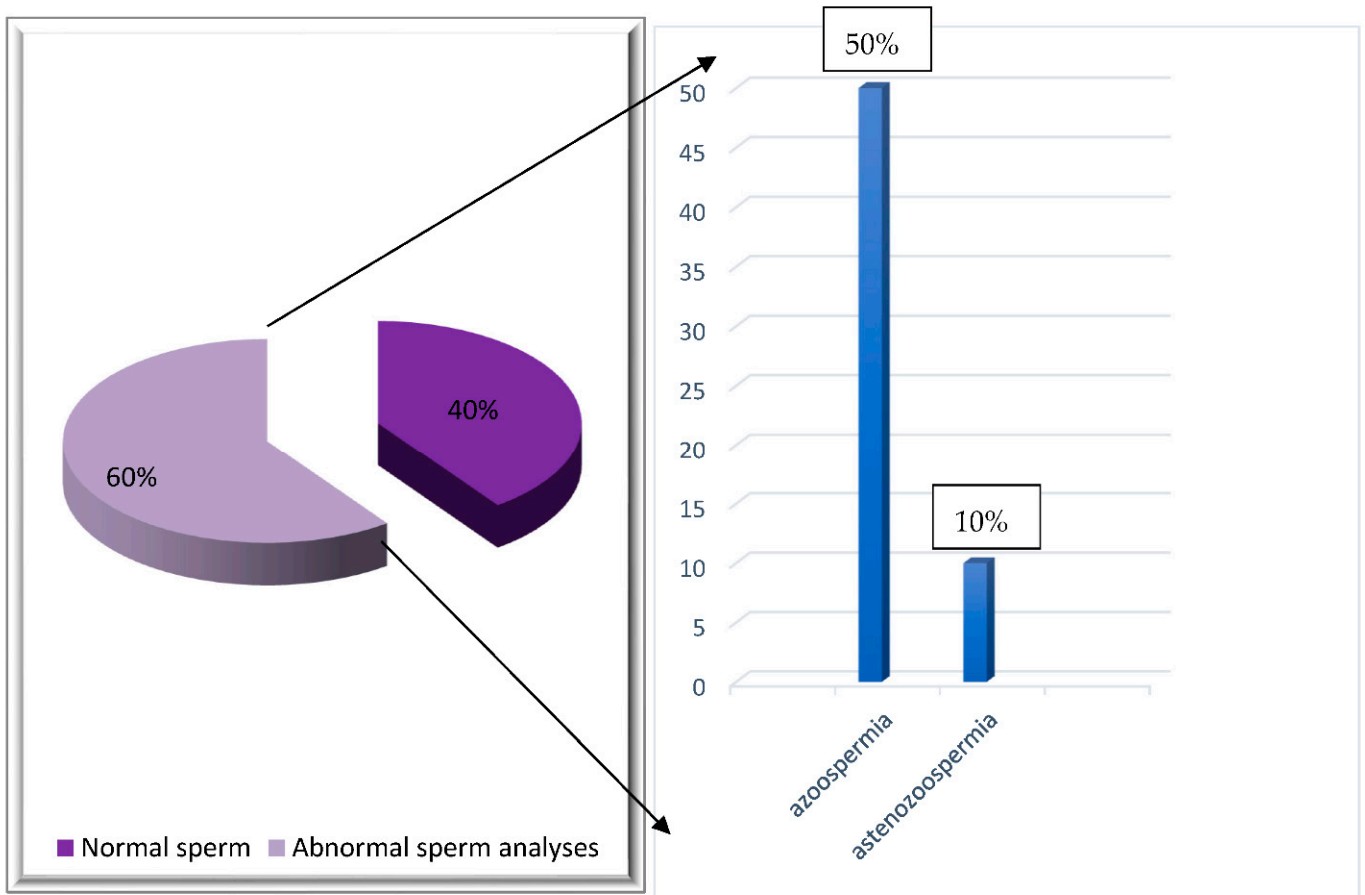

**Figure 3.** Distribution of HD patients according to spermogram findings following BEACOPP treatment.

Neither the post-ABVD nor the post-BEACOPP group showed that there are correlations between the age of patients and spermogram abnormalities.

*3.2. Female Survivors*

Within the studied cohort, full data following treatment was available for 81 patients, having a median age of 27.3 (range: 14–49) years. All patients stated regularity of their period prior to treatment.

According to disease extent, the evaluated female patients have the following distribution: 8 were with CS I, 30 with CS II, 15 in CS III and 28 with CS IV. The analyzed subgroup correlates with the overall incidence of female HD patients in different clinical stages of the disease, i.e., there are no significantly outstanding occurrences by clinical stage categories. Following treatment, the overall number of patients with infertility issues is insufficient for any kind of statistical correlation examination against distinct clinical stages of disease, or regarding limited and advanced disease as collective categories.

Following ABVD treatment, out of the total of 74 patients evaluated, 81.1% (60 pts) of the patients remained with a regular cycle, in 13.5% (10 pts) time irregularities in the cycle were reported and in 5.4% (4 pts) of the patients the cycles had ceased (Figure 4).

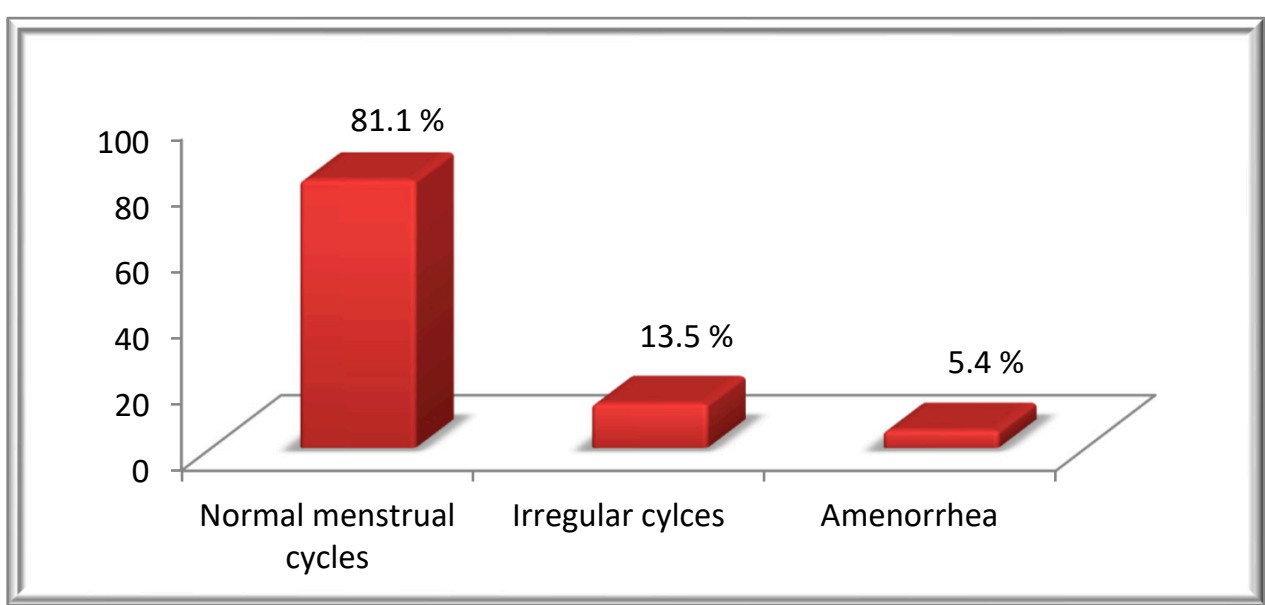

**Figure 4.** Distribution of HD patients according to menstrual status following ABVD treatment.

The analyses showed a statistically significant correlation between the age of patients (under or over 30 years) and the status of the menstrual cycle following ABVD chemotherapy (Pearson chi-square = 18.5496, *p* = 0.000094). Over 86% of the patients who developed irregularities in their cycle were above 30 years of age.

Seven female HD patients, treated with BEACOPP chemotherapy were analyzed, all of them stating that their periods were regular prior to treatment. Following BEACOPP therapy, 57.1% (4 pts) remained with a regular cycle, 14.3% (1 pt) developed irregularities regarding time (variations in occurrence and/or duration) and 28.6% (2 pts) reported cessation of the cycle (Figure 5).

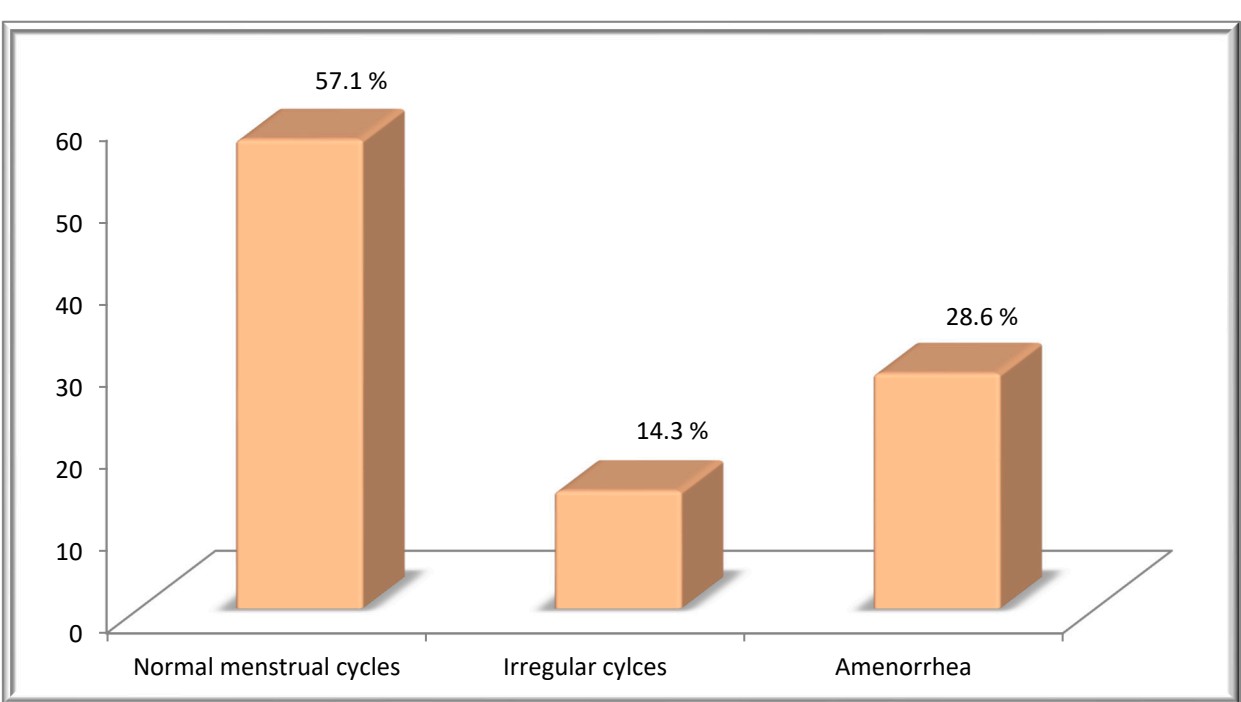

**Figure 5.** Distribution of HD patients according to menstrual status following BEACOPP treatment.

## 4. Discussion and Conclusions

The history of treating Hodgkin's lymphoma, in a quite traditional manner, defines three distinct periods: approach determined by disease characteristics, approach defined by the treatment option, and, since 1995, an approach oriented towards the patient. With undeniable respect, we acknowledge the immense and substantial research activities during the first two periods: investigating the etiology, definition and developing the variety of diagnostic procedures, improving and continuously enhancing the therapeutic methods and options. Nevertheless, the top priority of the contemporary approach in treatment during the last two or three decades has been achieving high quality of life for the patient, which is defining the period as patient oriented.

The progress achieved through treatment with aggressive, but optimized therapeutic regimens, essentially determines the HL survivors as one of the largest groups of long living cancer patients in the world. Even though it does leave open the issue of potential long-term adverse effects due to treatment, such as the risk of developing secondary malignancies, the risk of non-neoplastic complications, mainly cardiac and pulmonary, as well as the issue becoming of significant importance to the patients, the risk of developing secondary infertility.

Hematologists should be able to promptly recognize the onset of late treatment complications. Aside from the risk for developing secondary malignancies and non-neoplastic organ and system complications, as a component of the overall strategy for managing these patients, they should be prepared to address the issue of fertility as a potential complication in the initial stage of the process. It needs to be acknowledged by all of us that this issue represents a crucial concern for patients in their reproductive age, who are subjected to chemotherapy.

Existing studies suggest that there is a connection between HL and abnormalities in spermatogenesis prior to treatment [19,20]. Assessment of the sperm quality within the diagnostic process, meaning before administering any kind of therapy, in patients included in this study, documented abnormal findings in as high as 15% of them. This calls attention to the possibility that the disease itself influences the quality of the sperm. In our patient cohort, the abnormalities in spermatogenesis are associated with the B type (symptomatic) disease, as well as with elevated ESR, which could be attributed to the secreted cytokines (due to the systemic effect of the disease itself). Other variables, both constitutive and disease presenting, had been challenged regarding statistical impact over occurrence of infertility problems, but none manifested statistical significance. However, it needs to be pointed out again that the assessment of infertility issues in our study was performed with methods available and used in the respective time period, and that future utilization of more sophisticated and advanced methods may reveal a different insight. Nevertheless, clinical and scientific uncertainties remain regarding the pathophysiology of gonadal dysfunction associated with HL.

Regarding the fertility state in females, the situation is more complex. Aside from simply registering that the cycle has been restored, further functional evaluations are warranted. These include hormonal investigations as well as morpho-functional examinations with ultrasound techniques. Although determination of FSH levels seems most appropriate, normal values do not necessarily confirm functional ovarian capacity. Determination of the inhibin B hormone should be of similar significance, since its levels have a reverse proportional influence on FSH. A more contemporary finding is that the most informative assay is the determination of anti-Müllerian hormone levels, since it is proportional to the number of generated follicles and illustrates functionality. Transvaginal ultrasound detects the size of ovaries, as well as the number of follicles, both of which should confirm active fertility and existing reserves. In summary, established recommendations have not been agreed upon and contributions to this segment need to be derived from future research.

Within the studied patient population, patients treated with ABVD chemotherapy, both male and female, demonstrated a low infertility risk, confirming that ABVD, as a regimen not containing alkylating agents, bears a low infertility potential. The analysis

revealed that the age above 30 years before commencing induction chemotherapy in women significantly increases the late risk for infertility, since this rate is inversely proportional with the reserve status of the ovaries. In contrast to women, for male patients undergoing chemotherapy age has not been shown to be indicative of the risk for infertility, which could be explained by the fact that in men the spermatogonia reserves are invariable during life. Our reported results for patients following BEACOPP chemotherapy show considerable impairment of the fertility potential, with an incidence of 60% in male and 43% in female patients, data quite comparable to relevant studies in which the impact of this treatment approach on fertility status was investigated [21,22]. Not ignoring the fact that our patient cohort is too small for using it as a validation tool and lacking the necessary power level, we are still confident that we have succeeded in deriving and confirming a considerable quantum of reliable information from this analysis.

It is necessary that awareness among all chemotherapy prescribing physicians is escalated regarding the undesired adverse effects on the reproductive potential of the most frequently used treatment regimens, as well as regarding the available methods for fertility preservation. Coordinated multidisciplinary discussions encompassing hematologists, assisted reproductive technology teams, embryologists, researchers, social workers and surgeons are highly warranted, maintaining the patient in the very focus of their collective attention, effecting the goals of improving the presented quality of information, definition of the risk rate resulting from eventual treatment consequences, as well as securing qualified expert guidance regarding measures for fertility preservation tailored individually for every patient. Sperm and embryo cryopreservations are considered standard practice today, and are widely available, bur other methods are either in investigational stages or performed in specialized centers that possess the required level of expertise for this category of procedures.

Being involved in the field of hematological oncology for decades does change perspectives. For a very long time our primary concern was to enable life for (then) unpromising patients. The situation has drastically and positively changed in reality and now we need to switch to securing quality of life for this population.

**Author Contributions:** All authors were involved in the study design. All authors analyzed and interpreted the patient data. G.A. was a major contributor in writing the manuscript. O.K. and A.S. critically revised the paper. All authors have read and agreed to the published version of the manuscript.

**Funding:** There was no source of funding for this study.

**Institutional Review Board Statement:** The Ethics Committee of the University "Sts. Cyril and Methodius", Medical Faculty, Skopje, approved this study.

**Informed Consent Statement:** Informed consent was obtained from all individual participants included in the study in accordance with the Declaration of Helsinki.

**Data Availability Statement:** Data files available from authors, upon request.

**Acknowledgments:** The authors wish to thank Rosalinda Isjanovska, epidemiology and biostatistics, for her statistical programming support in the preparation of this manuscript and all the patients who participated in this study.

**Conflicts of Interest:** The authors do not have any conflict of interest.

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
