# Peer review of "Reproductive Issues in Long-Term Surviving Patients following Therapy for Hodgkin’s Disease in the Republic of North Macedonia: Risks of Infertility According to First-Line Treatment Regimens"

_hematolrep, doi:10.3390/hematolrep14020013_

Round 1

Reviewer 1 Report

Although the findings are interesting (not novel), the presentation and the analysis of the findings are not in an ucceptable level : the epidemiologic, clinical and laboratory characheristics of the patients must be presented clearly (preferentially in a table) and univariable and multivariable analysis performed for the characteristic studied. The follow up of patients regarding the fertility parametes and possible repeated analysis must be included. Advise from a statistician may be particularly helpful.

Figures for the BEACOPP group must be included and the level of the language must be considerably improved.

Author Response

Regarding suggestions from Reviewer 1:

  1. Language has been reviewed throughout the entire manuscript and adequate corrections/changes have been incorporated. One of our co-authors (O.K.) is a native English speaker by education (in the U.S.).
  2. The Introduction segment has been enriched and widened, addressing a broader range of fertility issues.
  3. Regarding the Research design, we can only state that this is a retrospective analysis of patients, mostly observational, supplemented by follow-up data. It is necessary to state that the manuscript has been submitted initially over one year ago (to the previous, Italian office), which is why the concept could not be extensively altered.
  4. The Methods used are listed following the statistical analyses, excluding those that were not adequate in terms of possibility to apply to small series, or those that reported "sample to small for comparative analysis". Only methods returning analytical results (regardless whether positive or negative) are employed in obtaining the results. Patient characteristics are presented, as requested, in a table form.
  5. The Results segment is supplemented with the requested figures regarding patients treated with the BEACOPP regimen.
  6. The Discussion and conclusions segment has been enriched with regard to more contemporary research reports in the field of fertility assessment.

Reviewer 2 Report

The paper is interesting and confirms the finding from other studies on this very important subject: fertility after cancer treatment. 
The introduction and discussion could be shortened. However, a discussion on the statement, that female fertility is not jeopardized may be moderated, as the authors only present data on menstrual cycle regularity. No endocrine profiles are presented. This should be noted in the discussion. Admittingly irregularity in the menstrual cycle  provides a hint to fertility. 

Author Response

Regarding suggestions from Reviewer 2:

  1. The language issue has been addressed, hopefully adequately, both in terms of spelling and grammar.
  2. In view of remarks posted by Reviewer 1, it was not possible to shorten the segments, but rather an expansion was requested. Therefore, we tried to explain the grounds for performing this kind of an analysis, and then to place our findings in perspective with research results and observations from peer institutions and experts, with the possibility that it would represent a contribution to this important contemporary subject. We enriched the discussion with respect to the stated remarks and suggestions, clearly addressing other existing possibilities of investigating, evaluating and predicting infertility issues, but hereby admitting that at the time of the analyses our institutions did not have the capacity for such advanced examinations (although even other authors admit that guidelines in this area are still not substantial).

Round 2

Reviewer 1 Report

The paper, although an obsevasion study, has been improved considerably. Additional issues include:

In the results sections the authors must add whether there is (or not) a corellation of fertility parameters with the stage of disease (if not adequate numbers available they should group the patients in 2 categories: advanced stage and not).

I did not see included the follow up information of the study goup in the M&M section

Author Response

Regarding suggestions from Reviewer 1:

  1. We included the analyses by clinical stage of disease in the "Results" section, both for the male, as well as for the female patients, with adequate interpretations, elaborations and explanations, as requested.
  2. We addressed the issue in the "Discussion and conclusions" section as well.
  3. We added the follow-up information and plan in the "Materials and methods" section, as requested.

We express our respect and gratitude for the efforts, suggestions and assistance, with hope that the modifications comply with the requests sent to us.

The authors